# Autoimmune Vestibulopathy—A Case Series

**DOI:** 10.3390/brainsci12030306

**Published:** 2022-02-24

**Authors:** Surangi Mendis, Nicola Longley, Simon Morley, George Korres, Diego Kaski

**Affiliations:** 1Department of Neuro-Otology, Royal National ENT and Eastman Dental Hospitals, UCLH, London WC1E 6DG, UK; surangi.mendis@nhs.net; 2Department of Neuro-Otology, St Ann’s Hospital, Whittington Health NHS Trust, London N15 3TH, UK; 3Hospital for Tropical Diseases, UCLH, London WC1E 6JB, UK; n.longley@nhs.net; 4London School of Hygiene and Tropical Medicine, University of London, London WC1E 7HT, UK; 5Department of Radiology, UCLH, London WC1E 6AS, UK; simon.morley3@nhs.net; 6Department of Neuro-Otology and Audiovestibular Medicine, Attikon, University of Athens, 15772 Athens, Greece; gfkorres@gmail.com; 7Department of Neuro-Otology, Leicester University Hospitals, Leicester LE2 7HL, UK; 8Centre for Vestibular and Behavioural Neuroscience, Department of Clinical and Movement Neurosciences, University College London, 33 Queen Square, London WC1N 3BG, UK

**Keywords:** AIED, autoimmune, vestibulopathy, vestibular, ulcerative colitis, Cogan syndrome, COVID-19, vaccination

## Abstract

Autoimmune inner ear disease (AIED) is a rare clinical entity. Its pathogenicity, heterogenous clinical presentation in the context of secondary systemic autoimmune disease and optimal treatment avenues remain poorly understood. Vestibular impairment occurring in the context of AIED is rarely subject to detailed investigation given that the auditory symptoms and their responsiveness to immunosuppression are the focus of the few proposed diagnostic criteria for AIED. We present three cases of vestibulopathy occurring in the context of autoimmune inner ear conditions, including the first known report of autoimmune inner ear pathology arising with a temporal association to administration of the Pfizer-BioNTech SARS-CoV2 vaccination. We review the available literature pertinent to each case and summarise the key learning points, highlighting the variable presentation of vestibular impairment in AIED.

## 1. Introduction

Autoimmune inner ear disease (AIED) is a disabling but rare autoinflammatory condition of the inner ear, often leading to progressive and fluctuating, sensorineural hearing loss. It was first described in 1979 with a case series authored by McCabe et al. [1]. Although more common in young and middle-aged women, as seen in many autoimmune conditions, it is heterogenous in its causative pathologies and clinical presentation (30% of cases are associated with another autoimmune condition [2,3,4]). This, in conjunction with the lack of established diagnostic criteria or guidelines for investigations, often leads to delays in recognising AIED and initiation of appropriate treatment.

The prevalence of AIED is estimated to be around 15 in 100,000 people (<1% of all cases of sensorineural hearing loss in the USA) [2] although this is likely to be substantially underestimated. In this article we outline the definitions, differential diagnoses and potential management options in three patients with suspected AIED.

There are currently no universally accepted diagnostic criteria for AIED, although several definitions have been proposed:

**AIED** is defined as progressive hearing loss (with or without vestibular dysfunction) arising secondary to autoimmune-driven inner ear pathology [5]. It is thought to occur due to autoantibodies directly targeting the inner ear or bystander damage resulting from circulating immune complexes [2,3], although the exact pathophysiological mechanisms underpinning AIED remain unclear. The most widely accepted presentation for diagnosis is that of bilateral, usually asymmetric, sensorineural hearing loss (SNHL), 30 dB HL or higher, rapidly progressing between 3 and 90 days [2,3,6,7] which is responsive to an immunomodulatory agent. Of note, vestibular impairment does not feature as a major criterion in any proposed classification so far [2,8,9]. Thus, by definition, vestibulopathy alone cannot occur as the sole cause of inner ear dysfunction in AIED.

**Primary AIED** comprises isolated inner ear dysfunction.

**Secondary AIED** describes inner ear dysfunction in the presence of systemic autoimmune disease (Table 1). The hearing loss, and vestibular involvement if this features, typically arises in the context of pre-existing autoimmune pathology, although occasionally the inner ear involvement can be the presenting feature of a new multi-organ autoimmune disorder.

Making a confident diagnosis of primary AIED is challenging, complicated by the absence of specific serological biomarkers. In the 1990s, various groups attempted to detect autoantibodies to inner ear components in those with suspected AIED. Harris and Sharp identified a 68 kD inner ear antigen, an antibody to Heat Shock Protein 70 (HSP70), on Western blot analysis of inner ear proteins using patient serum [12]. This antigen was thought to be present in one-third of all patients with suspected AIED, resulting in the now commercially available OTO-blot test. Early studies found a correlation between antibody-positive patients and steroid-responsiveness [13], although the relationship between HSP70 and AIED was later disputed [14] and HSP70 antibodies were also found in subjects with multiple systemic conditions including various malignancies, cardiovascular disease, and even healthy subjects [15,16], rendering the OTO-blot test less useful.

AIED research and clinical descriptions have largely focused on the hearing component of the inner ear. However, vestibular involvement is seen in approximately 50–60% of cases [2,6,17]. The reported vestibular symptoms in AIED are heterogeneous and include acute vertigo, disequilibrium, ataxia and positional vertigo [18,19]. Many case reports have not explicitly explored the cause of such symptoms, and vestibular function was not objectively evaluated. In this small case series, we highlight some of the vestibular presentations in AIED and propose that vestibulopathy should be actively pursued and formally investigated in all patients suspected to have this condition.

## 2. Cases

### 2.1. A: Case 1

A 28-year-old male carpenter of white-British origin was referred to our tertiary Centre in December 2020, a year after developing sudden-onset spinning vertigo, imbalance and bilateral hearing loss. He had a 10-year history of ulcerative colitis which was in clinical remission on Azathioprine 200 mg daily. He also had long-standing classical migraine.

He had a stormy clinical course over the first year; he was unable to work due to episodic vertigo, oscillopsia (due to primary gaze nystagmus) and increasing migraine frequency with new visual aura.

The hearing loss was initially asymmetric, predominantly low-frequency and rapidly fluctuant over several weeks. When he first presented locally, he was treated with multiple short courses of Prednisolone whenever his hearing deteriorated, usually resulting in marked rapid improvement. Without a clear, defined local or national management pathway for AIED, serial audiograms were not routinely performed and his treatment with oral steroids was variable (50–60 mg on a tapering regime, usually over 7–14 days). Intratympanic steroid administration was attempted to avoid long-term oral steroid use but hearing recovery was not seen with three, separate, early attempts at this.

Objective vestibular function tests conducted locally three months prior to referral to our Centre showed bilateral vestibular impairment. Contrast-enhanced magnetic resonance imaging (MRI) of the internal auditory meatus (IAM), in addition to a full auto-immune and vasculitic screen performed locally at the same time, were normal.

In June 2021, 18 months after his initial onset of symptoms, repeat focused vestibular examination and investigations at our Centre confirmed persistent bilateral vestibulopathy. The clinical findings and objective audiovestibular function tests are summarised in Table 2 and Table 3 and Figure 1. Delayed gadolinium contrast-enhanced MRI IAMs imaging is given in Figure 2.

Given the fluctuant, steroid-responsive hearing loss and vestibular involvement occurring in the context of pre-existing ulcerative colitis, a diagnosis of secondary AIED was made. He had a good initial response to high-dose oral prednisolone, although his hearing loss recurred on steroid withdrawal, requiring a maintenance dose of prednisolone 30 mg/day (administered with bone-protection and a proton pump inhibitor). Despite these measures, there was further deterioration of his baseline hearing over a two-year period. His ulcerative colitis remained quiescent during this time.

Alongside the pharmacological management, he engaged in self-directed vestibular rehabilitation and a regular exercise program (swimming and walking). This led to subjective improvement in his balance and oscillopsia.

He was managed as part of a multidisciplinary, multi-site team with additional input from rheumatology, gastroenterology and ENT with regards to the most appropriate steroid-sparing maintenance agent. A preliminary decision was made to consider Methotrexate, but other options included anti-TNFα therapy, or a dose increase of Azathioprine, and a final decision is awaited.

### 2.2. B: Case 1—Discussion

The subject in this case of AIED secondary to systemic ulcerative colitis presented with rapidly progressive, fluctuant, steroid-responsive hearing loss with episodic rotatory vertigo and imbalance secondary to bilateral vestibular impairment. His case demonstrates how central vestibular compensation occurs following a peripheral vestibular insult due to the plasticity of the vestibulo-ocular, vestibulo-spinal, vestibulo-collic and vestibulo-autonomic pathways, particularly in the young [23]. The objective vHIT findings are reflective of this dynamic vestibular compensation over time, demarcated by the emergence of covert saccades during head impulses. It is worth noting that the hearing loss of AIED was responsive to immunosuppression, at least initially, but that glucocorticoids are rarely used to treat peripheral vestibular loss given they do not appear to improve balance in the long term [24,25], even though use may result in acute symptomatic improvement [26].

Whilst the presence of an underlying autoimmune disorder supports AIED, a differential diagnosis would be bilateral Ménière’s disease, particularly as it is postulated that AIED and bilateral Ménière’s disease could be one single entity in continuum [27]. The imaging findings lend weight to this. Alternatively, the clinical history of migrainous headaches following the onset of his hearing loss could imply that the vestibular symptoms are also, at least in part, attributable to vestibular migraine. One would not expect the documented bilateral vestibular impairment seen in this case to occur secondary to vestibular migraine; however, it is certainly possible for migraine and AIED to co-exist. Multiple genetic and environmental factors are thought to be shared by headache, particularly migraine, and autoimmune conditions [28,29]. Consequently, immunological activity in the context of migraine may possibly prime some subjects to develop immune-mediated disease.

This case also highlights the difficulties encountered by both the patient and treating clinicians when a clear management pathway for a rare disorder does not exist. This is of specific importance in AIED where funding issues for expensive biologics and disease-modifying antirheumatic drugs (DMARDs) can hinder management of a time-sensitive clinical condition.

### 2.3. A: Case 2

A 17-year-old man of Ghanaian heritage was referred to the neuro-otology clinic at our Centre in June 2019 with a four-month history of insidious right-sided hearing-loss and unsteadiness of gait, followed later by sudden complete loss of hearing on the left and rapid deterioration of his balance.

The only significant past medical history was an isolated episode of bilateral uveitis at age four. This resolved at the time with Dexamethasone eye drops, which he continued long term with ophthalmology follow up.

Upon presentation to us he had profound, bilateral, sensorineural hearing loss. He also had bilateral vestibular impairment and a global absence of peripheral reflexes. He had no recurrence of eye disease whilst taking corticosteroid eye drops. A summary of the investigations performed between June 2019 and June 2021 at our Centre is given in Table 4 and Table 5 and the pure tone audiograms and vHIT traces are illustrated in Figure 3.

The hearing loss was initially treated with a left intra-tympanic steroid injection and oral Prednisolone 50 mg/day for six weeks. His hearing on the left improved significantly, particularly in the low frequencies.

Post-contrast MRI IAMs (Figure 4) and brain showed bilateral asymmetric cochlear enhancement and left-sided enhancement of the vestibule, lateral semicircular canal and cochlear nerve. Pre-contrast T1 shortening was consistent with possible Cogan syndrome but differentials of inflammatory conditions (sarcoid, connective tissue disease, Behçet’s disease), infection and CSF-borne spread of neoplastic disease were also considered.

A PET-CT showed widespread reactive lymphadenopathy. USS guided lymph node biopsy was not diagnostically helpful, although this was undertaken in the context of immunosuppression, secondary to steroid treatment. All other investigations were normal including nerve conduction studies, CSF analysis and an autoimmune screen.

A diagnosis of atypical Cogan syndrome, a rare autoimmune condition affecting the eye and inner ear, was made. The Dexamethasone eye drops were discontinued. After completing six weeks of oral Prednisolone 50 mg/day, this was tapered down by 10 mg per month with a proton pump inhibitor and bone-protection. Long-term maintenance on a steroid-sparing agent, Mycophenolate Mofetil (MMF) 1 g twice daily, followed upon advice from the neuro-immunology team.

His investigations and management followed multidisciplinary team advice with ENT, neuro-genetics, immunology, audiology and physiotherapy input. Although he saw significant improvement in his left-sided hearing loss and marginal improvement on the right, he went on to develop symptoms of visually induced vertigo, secondary to the underlying vestibulopathy. He was fitted with a left hearing aid and continues to benefit from vestibular rehabilitation.

### 2.4. B: Case 2—Discussion

Although Cogan syndrome (CS) is a condition consisting of audiovestibular dysfunction, the focus of most reports on the subject is hearing loss, not specifically vestibular involvement, as with most other AIEDs. This is despite vestibular function being found to be abnormal in up to 90% of subjects with CS [30]. This vestibular involvement typically consists of Ménière’s-like episodes with intense spinning vertigo and nausea, or vomiting associated with fluctuating aural symptoms [31,32,33]. However, the case we present was one of sudden onset bilateral vestibulopathy. As outlined in Case 1, treatment of bilateral vestibulopathy consists primarily of vestibular rehabilitation to enhance central vestibular compensation in the absence of peripheral recovery. When bilateral vestibulopathy presents suddenly in a young individual, autoimmune and genetic aetiologies should be considered.

In the authors’ experience, most clinicians associate peripheral vestibular impairment with episodic vertigo but it is less well recognised that bilateral, sudden onset dysfunction can present with isolated marked imbalance, ataxia and oscillopsia in the absence of any reported dizziness or vertigo [34]. Patients may report a ‘staggering’ gait. Bilateral vestibulopathy can be confirmed clinically by the presence of bilateral refixation saccades on lateral, and possibly vertical, head thrust testing, and by marked impairment of static and dynamic balance (Romberg testing and tandem gait, respectively) [35]. A drop of two lines or more on testing of dynamic visual acuity is also suggestive [35].

CS is an autoimmune vasculitis resulting in audiovestibular dysfunction and ocular inflammation, the latter most commonly manifesting as interstitial keratitis. Two distinct groups exist: those with typical or atypical disease. Typical CS occurs in young adults with flares of interstitial keratitis and Ménière’s disease-like vertiginous episodes resulting in progressive sensorineural hearing loss within approximately two years [30,36]. In the atypical form, children are more likely to be affected, the ocular complications may include flares of uveitis, conjunctivitis or episcleritis, and the onset of the audiovestibular impairment and ocular complications can be separated by a much longer timescale of up to 11 years [30,36,37], or perhaps longer as illustrated by this case. The eye is affected first in 41% of cases, the ear is affected first in 43% and both organs are affected simultaneously in 16% [38].

CS is rare, with just over 300 cases reported in the literature [36], although this figure is not reflective of the true incidence given that many cases likely remain unrecognised and therefore undiagnosed. First described in 1934 by Morgan and Baumgartner as a case of ‘Ménière’s disease with interstitial keratitis’ [39], typical CS most commonly affects adults in their third decade with no gender predilection [30,31], although earlier onset, atypical disease is seen more commonly in males [36]. The literature available to date suggests that the incidence of CS is highest in Caucasians [31].

Following the discovery of antibodies to inner ear and corneal antigens in patients suspected to have the condition, and given its favourable response to immunomodulatory drugs, CS was originally defined as systemic autoimmune disorder under the umbrella of AIED. There is no diagnostic test or pathognomonic marker and the diagnosis is therefore a clinical one [40].

Corticosteroids remain the initial treatment modality, with the primary aim of preventing progressive irreversible audiovestibular loss and ocular inflammation or complications from secondary vasculitis. In a study of 62 patients, 84% responded to oral steroids in isolation, 90% showed improvement with DMARD therapy and 100% responded to biologics [41]. When a patient’s steroid-responsiveness wanes over time or if steroid-dependency develops, Infliximab in combination with oral steroids is emerging as a promising, efficacious treatment option, although success is also reported with various other DMARDs including methotrexate, mycophenolate mofetil and azathioprine [30].

As with Case 1, we also highlight the importance of cross-specialty collaboration.

### 2.5. A: Case 3

A 46-year-old medical doctor of white-British origin had her first dose of the Pfizer-BioNTech SARS-CoV2 vaccination. That evening she experienced new-onset continuous, bilateral tinnitus. This continued and after two weeks a mild-to-moderate bilateral high frequency sensorineural hearing loss, which was worse on the left, was documented via her local ENT service. Apart from being atopic, she had no significant past medical history and she had not had clinical SARS-CoV2 infection itself.

A course of high-dose oral Prednisolone (40 mg for three days, increased to 60 mg for four days) was commenced with a plan to reduce this slowly over a further ten days but on day 12 she developed right ear erythema, pain, swelling and vesicular lesions. There was no history of fever or a systemic inflammatory response. Unfortunately, a possible viral aetiology was not investigated and no anti-viral treatment was given. She was treated with almost three weeks of antibiotics (a combination of one week of intravenous ceftriaxone, 10 days of oral ciprofloxacin and subsequently topical chloramphenicol) for suspected perichondritis. Complete resolution of the right ear symptoms was eventually seen.

Following this episode, repeat audiometry showed improvement in the hearing thresholds on the on the left but deterioration on the right (the side with the perichondritis). This also improved over time. Then, one month after the original vaccination, she suddenly experienced marked unsteadiness and head movement-induced oscillopsia without vertigo. vHIT shortly after this showed reduced vestibulo-ocular reflex (VOR) gain and overt refixation saccades bilaterally, more so on the right. vHIT repeated three months later showed reduced gain on the right once again but also both covert and overt catch-up saccades. Saccades were also present on Suppression Head IMPulse (SHIMP) testing at that time, indicative of ongoing central compensation.

Subsequent serial audiograms revealed bilateral high-frequency mild hearing loss; this was initially fluctuant on the right, but the thresholds became persistently raised over time. Additional investigations provided further evidence of right-sided audiovestibular dysfunction (Figure 5), absent Transient Evoked Otoacoustic Emissions (TEOAEs) on the right (not shown), complete canal paresis on the right on caloric testing, absent auditory brainstem response wave II on the right (not shown), and an abnormal cVEMP (reduced amplitude and later complete absence) on the right.

vHIT repeated several weeks later eventually showed bilaterally reduced VOR gain suggestive of bilateral vestibular dysfunction. A summary of the audiovestibular and additional investigations undertaken for this patient is given in Table 6 and Table 7.

The peripheral vestibular impairment did not fluctuate on objective testing and she remained constantly unsteady, requiring walking aids. She reported a one-off brief episode of rotational vertigo whilst turning in bed and some visual motion sensitivity at work, particularly after prolonged computer screen use. With time, central compensation facilitated by vestibular rehabilitation took place and her balance improved both subjectively and objectively (now walking without the need for walking aids).

This patient’s symptoms were extremely disabling, and she was concerned about the possibility of further vaccine doses resulting in progressive symptoms. Her age, occupation and the fact that she was a mother to two young children meant it was impossible for her to shield from symptomatic COVID-19 infection in the long term. Following discussion with the infectious diseases and virology teams, she received two further mRNA COVID-19 vaccination doses but from a different manufacturer, Moderna with Valaciclovir cover and serial audiovestibular testing. She had no further systemic or neurological symptoms and no adverse effects were seen apart from slight increased tinnitus perception.

The clinical picture was suggestive of an autoimmune inner ear disease, *possibly* triggered by the vaccine on a primed immune system. She continues to be closely monitored by the audiology, vestibular neurology, vestibular physiotherapy, rheumatology and infectious diseases teams.

### 2.6. B: Case 3—Discussion

In Case 3, the subject’s symptoms, including perichondritis, seemed to occur in a somewhat protracted, fluctuant manner following vaccination. However, although the symptom onset was temporarily associated with the first dose of the Pfizer BioNTech SARS-CoV2 vaccine, it is impossible to comment on true causality. As of 10th December 2021, 243 cases of ‘deafness’ following administration of the Pfizer vaccination have been reported via the UK MHRA Yellow Card reporting scheme, including 47 cases of sudden onset hearing loss. There have been 2011 reports of tinnitus and 1410 reports of vertigo [42].

In this case, the initial evolution of right-ear perichondritis with vesicles, pinna inflammation and a limited response to antibiotics may have been suggestive of a viral aetiology, potentially VZV, particularly as some cases of herpes zoster or herpes simplex reactivation associated with Pfizer BioNTech vaccination [43], and following clinically symptomatic COVID-19 infection [44,45], have been reported. Although the true cause is elusive, the clinical evolution of fluctuant, persistent, audiovestibular dysfunction would be in keeping with an autoimmune cause.

To date, there is only one report of vestibular neuritis occurring following COVID-19 vaccination [46] and a handful of case reports of vestibular neuritis arising in the context of active COVID-19 infection [47,48,49,50,51], the pathophysiological mechanism of the latter presumably involving reactivation of a latent herpes virus in the vestibular labyrinth or the Scarpa ganglion. Neurological complications have recently been shown to be present in up to 30% of patients with COVID-19 infection, with balance disorders affecting 18% [52].

The question remains as to whether vaccines or their adjuvants may trigger autoimmune disease. Such events can arise as a result of interaction between a susceptible, immunologically ‘primed’ subject in combination with reaction to a single or combination of vaccine components, in theory due to the concept of molecular mimicry [53]. Molecular mimicry refers to autoimmunity in a genetically susceptible individual being induced by exposure to a foreign antigen, possibly a vaccine adjuvant, acting as a cross-reactive epitope [54,55]. This can occur due to the adjuvant’s structural similarity to specific human proteins. Although it has been shown that immune autoreactivity can occur following vaccination in this way [56], it should be emphasised that such reactions are thought to be extremely rare and should not preclude vaccination or fuel vaccine hesitancy arguments. A recent non-systematic review detailing a practical approach for delivery of vaccinations in those already diagnosed with an autoimmune disease concluded that such patients should continue to receive vaccinations, ideally during low disease activity and when there is no intercurrent infection [57]. Vaccination can take place in the context of low-level immunosuppression without negatively impacting vaccination antibody responsiveness, although vaccinations should preferentially be administered before DMARD agents are initiated [57].

Case reports are emerging of SARS-CoV-2 vaccination being temporally associated with the onset of neurological autoimmune disease such as Guillain–Barre syndrome and Acute Disseminated Encephalomyelitis (ADEM) [58,59,60]. However, this is the first description of autoimmune inner ear disease occurring after COVID-19 vaccination to our knowledge.

## 3. Discussion

The three cases presented in this series highlight the variable presentation of vestibular involvement in autoimmune-related inner ear dysfunction.

The vestibulopathy and other baseline characteristics in these three cases are summarised in Table 8. Even within this small case series, we illustrate that the heterogeneity of vestibular impairment in AIED with gradual deterioration in balance function and episodic vertigo (Case 1) and sudden onset disabling vestibular dysfunction, which can be bilateral (Cases 2 and 3). Two of the three cases highlight that vestibular impairment can occur with sudden onset ataxia, imbalance and oscillopsia *in the absence of vertigo or dizziness*, which is crucial for non-vestibular specialists to appreciate to facilitate prompt referral to neuro-otology, diagnosis and treatment in this rare group of patients.

Vestibular involvement is seen in approximately 50–60% of cases of AIED [2,6,17] and given this relatively high figure it seems unusual that vestibulopathy does not feature as a separate criterion in any of the few existing diagnostic classification systems. We suggest that any future proposed diagnostic pathways should encompass this important feature, to improve of our understanding of the true incidence of vestibulopathy in both primary AIED and the various secondary AIED subsets outlined in Table 1. The absence of predefined management guidelines or pathways for AIED led to a lack of standardised investigations in these three cases herein described. We are developing a department-specific AIED algorithm to address this shortfall, with a view to national and international validation. vHIT was the vestibular function test of choice in all three cases in this series but, as seen in Case 3, a small battery of vestibular investigations may better evaluate the VOR at different frequencies (e.g., high frequency 4–7 Hz vHIT vs. low frequency 0.002–0.004 Hz caloric) and facilitate anatomical localisation of a peripheral vestibular lesion (evaluation of saccular, utricle and semicircular canal function or superior vs. inferior vestibular nerve involvement). Impulsive rotation and caloric tests are the more ‘traditional’ vestibular investigations but cVEMPs are increasingly used to non-invasively evaluate labyrinthine function, specifically that of the saccule and inferior vestibular nerve, in cases of moderate to profound sensorineural hearing loss [61].

Standardised diagnostic criteria for AIED are needed, as is a platform to prospectively collect data across multiple sites. This would highlight gaps in our collective understanding of AIED, illustrate best practice, and allow the neuro-otology community to move towards creating a usable treatment guideline and undertake future prospective collaborative work to gain international expertise and standardisation. Given the potential overlap between bilateral Ménière’s disease and AIED, the newly formed UK Ménière’s Disease Registry [62] will likely be ideal in this regard. A multitude of collaborative platforms have been created during the COVID-19 pandemic, for example, Accelerating COVID-19 Research and Development (ACCORD) [63] and similar models could be adopted in AIED specifically, particularly as low incidence pathology requires multi-site input to produce meaningful datasets.

Although no specific biomarkers exist in AIED, the newly developed MRI protocols involving delayed uptake of intravenous gadolinium show promising potential as a complementary diagnostic tool in AIED [64,65]. Whereas plain non-contrast MRI IAM scans are expected to be normal, gadolinium-enhanced three-dimensional fluid attenuation inversion recovery (FLAIR) imaging showing abnormal bilateral enhancement in the inner ears of patients with AIED has been reported in patients with (clinically) definite Ménière’s disease [66]. *Intratympanic* gadolinium three-dimensional MRI imaging has also been utilised to diagnose endolymphatic hydrops in AIED [67]. Although the use of this or IV gadolinium in MRI IAM imaging is not widespread or routine, radiological markers may be promising in the detection of early autoimmune ear disease.

The optimal treatment approach in AIED remains unclear at present. First-line treatment options for idiopathic sudden-onset sensorineural hearing loss include oral steroids, intra-tympanic steroid injections or a combination of both. Of these, the most effective treatment modality remains unclear but the three-arm UK STeroid Administration Routes For Idiopathic Sudden sensorineural Hearing (STARFISH) trial [68] due to commence in 2022 aims to address this. Similarly, the optimal length of treatment with oral steroids or frequency of intra-tympanic steroid injection in early or suspected AIED specifically is subject to debate and significant inter-clinician variability exists. At present, treatment is therefore decided largely on a case-by-case basis with steroid courses being used for an extended period, usually a month at a time, or longer, in an attempt to induce symptom remission. With regards to vestibular impairment specifically, although reversibility with steroid use is not seen in the same way as with hearing loss, it is unclear whether specific immunomodulatory drugs (oral or intra-tympanic steroids or alternative immunosuppressive agents) halt the progression of vestibulopathy in AIED and, if so, what specific dose or length of treatment is required. In cases where hearing deteriorates despite immunomodulatory therapy, cochlear implantation is a viable option often resulting in excellent outcomes [69], although ossification can affect up to 50% of patients receiving long-term treatment for AIED [70], and therefore early implantation is desirable to avoid traumatic insertion associated with post-inflammatory cochlear obliterative changes. Rubenstein et al. recently described generation of vestibular electrically evoked compound action potentials following placement of a vestibular-cochlear implant in two patients with bilateral Ménière’s disease and bilateral vestibular loss [71], and therefore vestibular-cochlear implantation may also emerge as a potential treatment option for AIED in the future.

Given that no diagnostic or treatment guideline is widely utilised at present, we propose that the next logical step would be to create a combined pathway to standardise data collection in AIED and develop a wider evidence base for management. We recommend that a full vestibular assessment (enquiry, clinical examination and vestibular function testing) be undertaken in all patients suspected to have AIED.

We believe that effective management of patients with such rare disorders requires close collaboration and inter-disciplinary advice across healthcare professionals (audiologists, vestibular physiotherapists) and specialties (ENT, rheumatology, immunology, infectious diseases, neurology).

## 4. Conclusions

The three cases discussed represent a wide spectrum of vestibular disease occurring in immune-mediated inner ear dysfunction, which has received little clinical focus in the literature to date.

We present the first known case of autoimmune inner ear dysfunction following administration of the Pfizer BioNTech SARS-CoV2 vaccination, although further reports and research are required to prove causality over a mere temporal association. Autoinflammatory symptoms following COVID-19 infection have been noted and it can be hypothesized that, by the same mechanisms, vaccinations or their adjuvants may also trigger an autoimmune response, perhaps in a pre-primed subject. However, this appears to be an exceptionally rare occurrence and should not, based on the evidence available to date, preclude subjects even with pre-existing autoimmune disease from receiving vaccines, particularly in the context of an ongoing global pandemic. It remains to be seen whether further reports of either COVID-19 infection or vaccination triggering autoimmune cochleo-vestibular disease come to the fore.

There is great potential for the nature of vestibulopathy in AIED, particularly in secondary AIED, to be characterised in further detail and for its optimal treatment to be understood. For this to take place, formal evidenced-based diagnostic criteria incorporating a vestibular assessment process should be agreed upon, which in turn requires a collaborative platform of audiovestibular, neurology and ENT professionals to gather data and compare outcomes from multiple sites.

## Figures and Tables

**Figure 1 brainsci-12-00306-f001:**
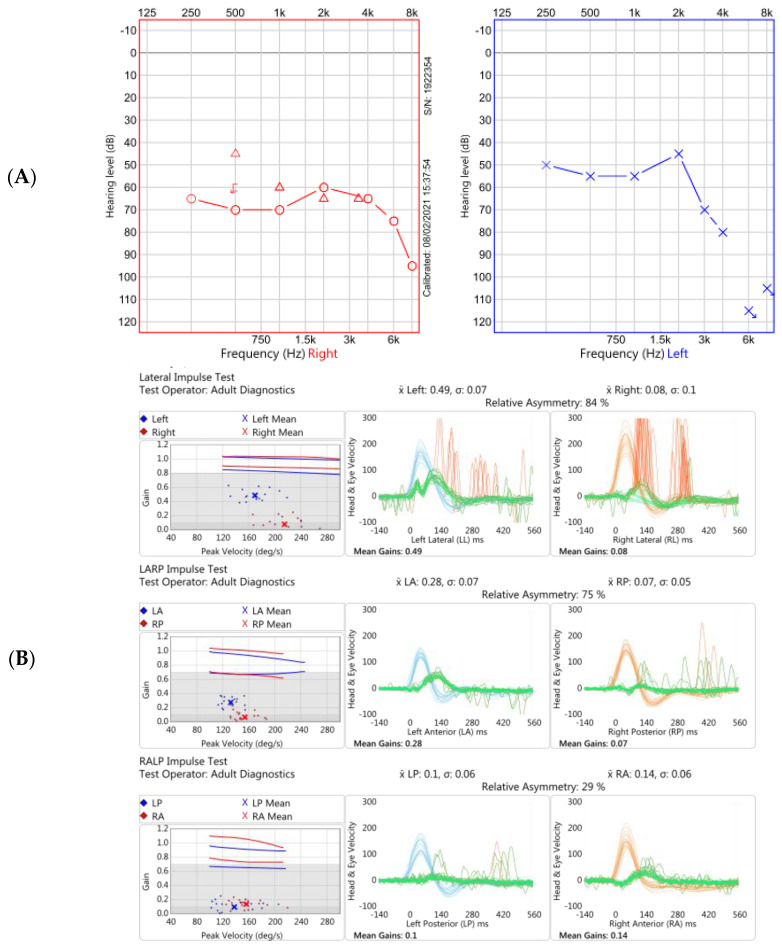
Two of the audiovestibular investigations performed at our Centre 18 months after original onset of symptoms. (**A**). Pure tone audiogram demonstrating moderate-to-severe hearing loss on the right (red trace) and moderate-to-profound loss on the left (blue trace). (**B**). Video head impulse test (vHIT) in Case 1 showing reduced gain (eye velocity: head velocity) of the vestibular ocular reflex across horizontal (top panel), and vertical (RALP plane, middle panel; LARP, bottom panel) planes. Note normal vestibulo-ocular reflex (VOR) gains using this vHIT device (GN Otometrics, Taastrup, Denmark); ≥0.8 for horizontal planes and ≥0.7 for vertical planes [20]. The traces show emergence of covert saccades occurring during the head movements, particularly on the right, reflective of dynamic vestibular compensation. Data not shown from impulsive rotational testing and oculomotor examination via videonystagmography.

**Figure 2 brainsci-12-00306-f002:**
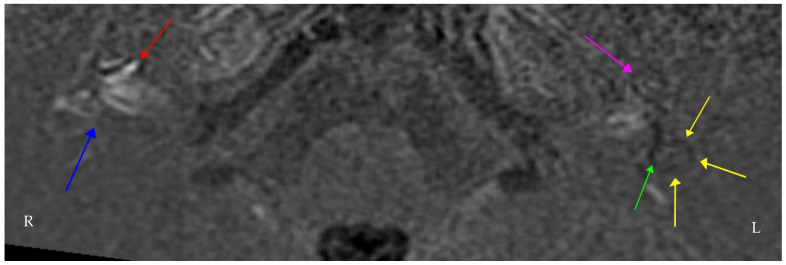
Axial Real Inversion Recovery (IR) MRI image 4 h post-intravenous gadolinium showing asymmetric increased enhancement of the right cochlea. The (**right side**) shows cochlear (red) and vestibular (blue) endolymphatic hydrops. The (**left side)** also shows cochlear (purple) and vestibular (green) endolymphatic hydrops extending into the lateral semicircular canal (yellow). Many centres worldwide have adopted similar protocols in delayed contrast-enhanced 3D FLAIR MRI imaging for suspected endolymphatic hydrops [21,22].

**Figure 3 brainsci-12-00306-f003:**
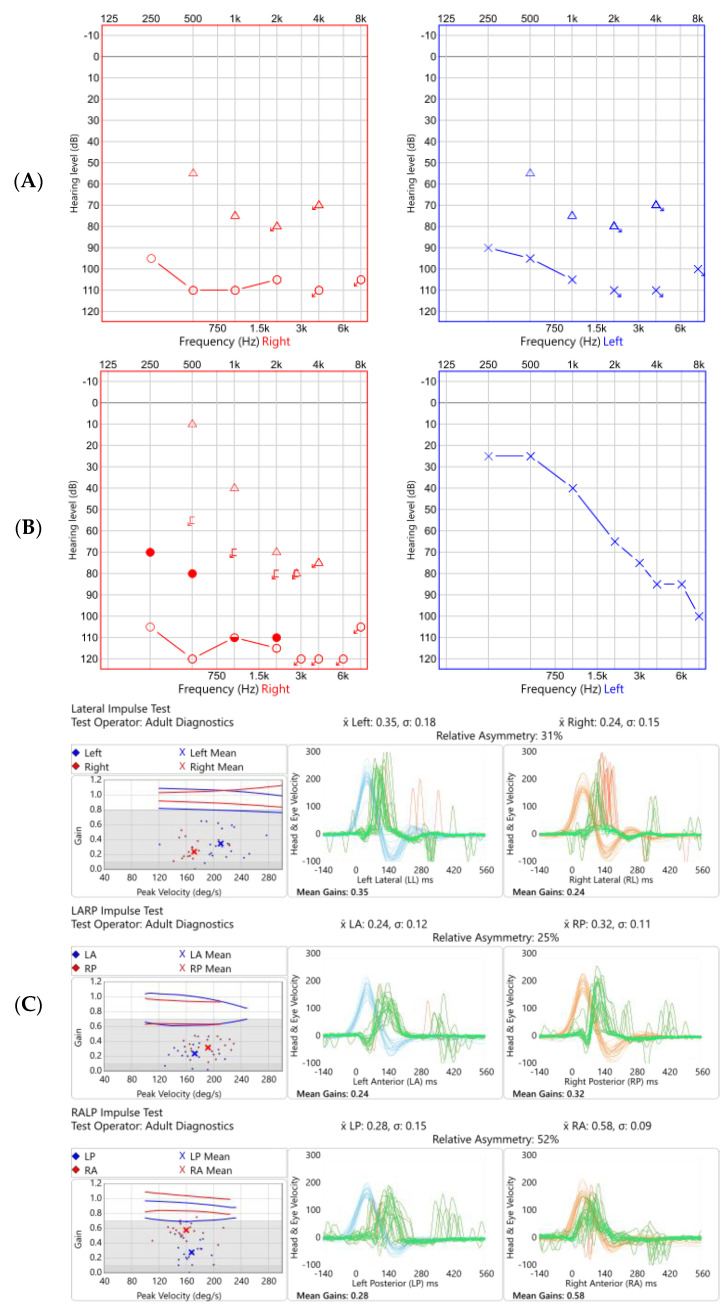
Audiogram and vHIT undertaken at initial presentation (**A**,**C** respectively) and almost two years later (**B**,**D**) in Case 2. Audiometry showed a bilateral profound hearing loss initially, which improved markedly on the left with time, particularly in the low frequencies. vHIT showed globally reduced gain (normal being ≥0.8 in the lateral plane, ≥0.7 in the vertical planes) in all six semicircular canals tested which marginally improved over time.

**Figure 4 brainsci-12-00306-f004:**
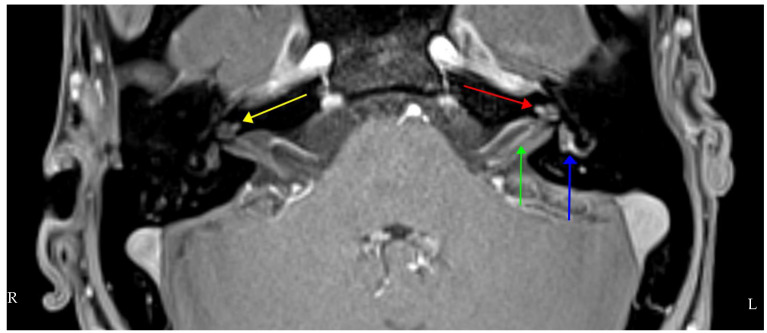
Axial T1 volumetric interpolated brain examination (VIBE) fat-suppressed (FS) MRI post-intravenous gadolinium showing bilateral cochlear enhancement (yellow **right**, red **left**), more on the left than the right and left-sided vestibular and lateral semicircular canal enhancement (blue). The left intracanalicular cochlear nerve shows increased contrast enhancement (green).

**Figure 5 brainsci-12-00306-f005:**
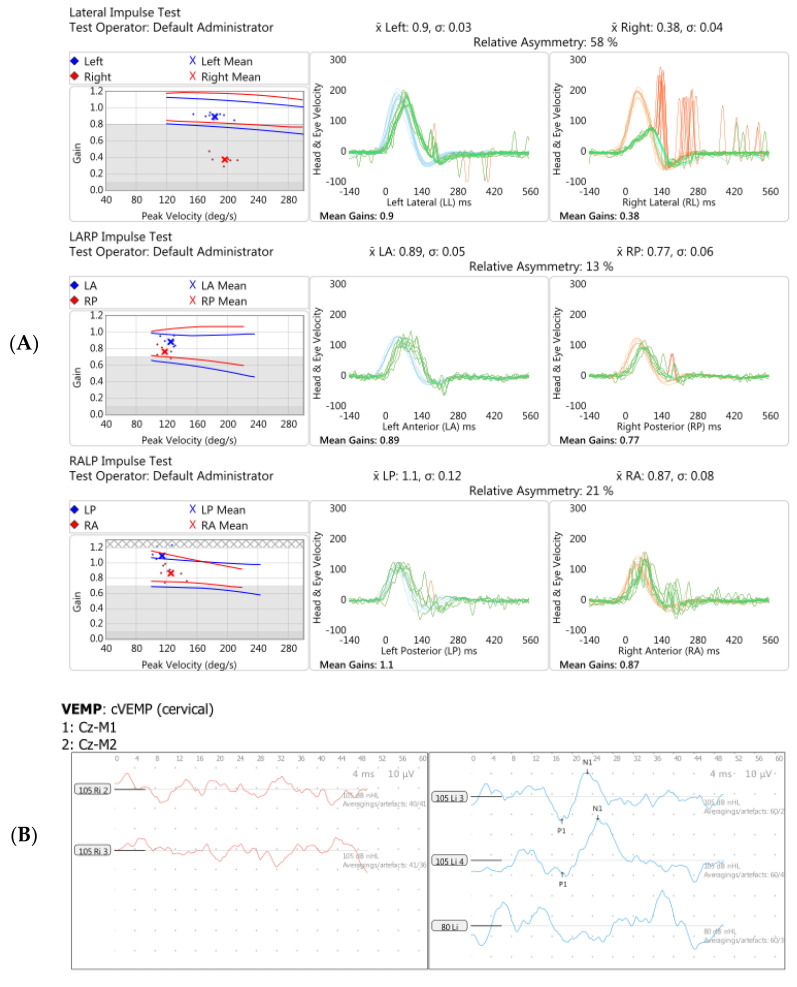
Vestibular investigations undertaken five months following Pfizer-BioNTech SARS-CoV2 vaccination after the onset of imbalance and ataxia in Case 3. (**A**). vHIT showed reduced gain (0.38) with refixation saccades in the right lateral plane. (**B**). Cervical Vestibular Evoked Myogenic Potential (cVEMP) response was absent on the right (red traces, left panel). (**C**). Monothermic caloric testing demonstrated 100% right-side canal paresis (note almost absent nystagmus on right warm and right cool traces).

**Table 1 brainsci-12-00306-t001:** Systemic autoimmune diseases seen with AIED.

Autoimmune Condition Associated with Secondary AIED	Prevalence of Hearing Loss	Prevalence of Vestibulopathy
Sjögren’s syndrome	46%	Unknown
Relapsing polytonicities	46%	Unknown
Behçet’s syndrome	30–63%	Orthostatic disequilibrium in 50% but prevalence of vestibulopathy is unknown.
Rheumatoid arthritis	25–72%	Unknown
Vogt–Koyanagi–Harada disease	49–51%	Unknown
Cogan syndrome	31–41%	90%
Systemic lupus erythematosus	6–70%	Unknown
Giant cell arteritis	7–100%	Unknown
Wegener’s granulomatosis	8–63%	Unknown
Mixed cryoglobinaemia	22%	Unknown
Ulcerative colitis	2%	Unknown
Systemic sclerosis	Unknown	Unknown
Antiphospholipid syndrome	Unknown	Unknown
Polyarteritis nodosa	Unknown	Unknown
Pyoderma gangrenosum	Unknown	Unknown
Takayasu’s arteritis	Unknown	Unknown
Hashimoto’s thyroiditis	Unknown	Unknown

Adapted from Mancini et al. 2018 [10], and Ralli et al. 2018 [11].

**Table 2 brainsci-12-00306-t002:** Summary of salient clinical examination findings and audiovestibular testing undertaken in June 2021.

Clinical Examination	Audiovestibular Testing	Imaging
Subtle left-beating nystagmus on left gaze only Bilateral catch up saccades in the horizontal plane on clinical head thrust testing (bilateral loss of vestibular function). Positive Romberg test Ataxic gait Normal positional (Dix-Hallpike) testing **Summary: bilateral vestibular impairment, more marked on the right. Evidence of impaired dynamic and static balance.**	Pure tone audiogram: bilateral moderate-to-severe SNHL, worse on the left Vestibular testing: Video head impulse test: bilateral low gain in both horizontal and vertical canals, with prominent covert and overt catch-up saccades on horizontal canal testing Impulsive rotational testing; minimal nystagmus generated bilaterally. **Summary: bilateral audiovestibular dysfunction with hearing marginally worse on the left and vestibulopathy worse on the right.**	Contrast-enhanced MRI four hours after IV Gadolinium: right cochlea enhancement, bilateral cochlear and vestibular hydrops and additional hydrops in the left lateral semicircular canal.

**Table 3 brainsci-12-00306-t003:** Summary of non-audiovestibular investigations undertaken between April–August 2021.

	Examination	Blood Tests	Other
Rheumatological	No rash	CRP, ESR: normal	
	No synovitis	ANCA, ANA, ENA: negative	
		Ro, La, * SM, * RNP, *S*CL-70, JO-1: negative	
		Centromere antibodies: negative	
		IgG and IgM anti-Anticardiolipin: negative	
		IgG and IgM anti-Beta-2-glycoprotein: negative	
		DRVVT	
Neurological	Peripheral and cranial nerve examination: normal	Purkinje cells: negative	


		Anti-Tr antibodies (screen): negative	
		Other cerebellar cells: negative	
		IgG white matter (myelin): negative	
		Anti-Hu antibodies: negative	
		Anti-Yo antibodies: negative	
		Anti-Ri antibodies: negative	
		MPO ELISA: normal	
		PR3 ELISA: normal	
Other	HR 70 regular, BP 138/84.	FBC, U&Es, LFTs, bone profile, thyroid function: normal	Urine analysis: normal
	Heart sounds: normal	Syphilis, HIV, hepatitis B serology, hepatitis C serology: not detected	
	Chest: clear.		

* SM = smooth muscle, RNP = ribonucleoprotein.

**Table 4 brainsci-12-00306-t004:** Summary of the salient clinical examination findings and audiovestibular investigations undertaken at presentation to our Centre in summer 2021.

Clinical Examination	Audiovestibular Testing	Imaging
No spontaneous- or gaze-evoked nystagmus, normal saccades and normal smooth pursuit. Bilateral catch-up saccades in the horizontal plane, more obvious on the right on head thrust testing. Positive Romberg test Ataxic gait–marked Impaired tandem gait Positional (Dix-Hallpike) testing: normal Remaining neurological examination–normal apart from *global peripheral hyporeflexia.* **Summary: bilateral vestibular impairment, more so on the left. Evidence of impaired dynamic and static balance.**	Pure tone audiogram: bilateral profound SNHL Otoacoustic emissions: bilaterally absent Acoustic reflexes: bilaterally absent Vestibular testing–video head impulse test: bilateral low gain in both horizontal and vertical canals, more so on the right, with prominent covert catch-up saccades on horizontal canal testing on the right **Summary: bilateral, largely symmetric, audiovestibular dysfunction.**	Contrast-enhanced MRI IAMs and brain: bilateral cochlear enhancement and enhancement of the vestibule, lateral semicircular canal and cochlear nerve on the left. Pre-contrast T1 shortening suggestive of Cogan’s syndrome but differentials include inflammatory conditions (sarcoid, connective tissue disease, Behçet’s disease), infection and CSF-borne spread of neoplastic disease. CT temporal bones: nil significant findings

**Table 5 brainsci-12-00306-t005:** Summary of non-audiovestibular examination findings and investigations undertaken at RNENT between June 2019 and June 2021.

	Examination	Blood Tests	Imaging	Other
Rheumatological	No rash.		Chest X-ray: normal	
	No lymphadenopathy			
	No synovitis.			
Neurological				Nerve conduction studies: normal


				Lumbar puncture and CSF analysis: normal



other	Heart sounds normal.			
	Chest clear.	FBC, U&Es, LFTs, bone profile, glucose, CRP: normal	Whole body FDG PET *:	US-guided axillary
	Abdomen soft and non-tender			lymph node
				biopsy: normal
	No palpable organomegaly		Widespread Lymphadenopathy. Repeat imaging showed mildly avid symmetrical axillary, pelvic-inguino- femoral lymph nodes. Reactive appearance, rather than pathological.	
				Urine analysis:
				normal










			USS abdomen: Lymphadenopathy within the abdomen and pelvis is comparable to the PET scan	

* Whole body *fluorodeoxyglucose* positron emission tomography.

**Table 6 brainsci-12-00306-t006:** Summary of clinical examination findings and audiovestibular investigations undertaken at presentation to our Centre between February and July 2021.

Clinical Examination	Audiovestibular Testing	Imaging
Gaze testing: normal Right-sided catch-up saccades in the horizontal plane on head thrust testing. Positive Romberg test Ataxic gait Positional (Dix-Hallpike) testing: normal Remaining neurological examination: normal **Summary:** **Right-sided vestibular impairment, evidence of impaired dynamic and static balance**.	Pure tone audiogram (serial): Initially—bilateral fluctuant high frequency sensorineural hearing loss, worse on left initially, then worse on right with perichondritis. Later—bilateral high frequency hearing loss. Audiovestibular testing: vHIT—reduced gain and catch up saccades bilaterally, worse on right. TEOAEs—absent on right. Caloric testing—canal paresis on right. ABR—absence wave II on right. cVEMP—reduced amplitude and later absence on right. **Summary:** **Left-sided, then right-sided high frequency hearing loss.** **Bilateral vestibulopathy, worse on the right.**	MRI IAMs and head, with and without Gadolinium contrast: normal

**Table 7 brainsci-12-00306-t007:** Summary of non-audiovestibular investigations undertaken at our Centre and other centres between February 2021 and July 2021.

	Blood Tests
Rheumatological	Cardiolipin IgG: (weakly positive).
	Lupus anticoagulant: normal
	ANA, ANCA, ENA, C3, C4: normal
	ESR, CRP: normal
	ACE: normal
	collagen type 2 antibodies: negative
	*HLA-B27 antibodies: positive*
	anti-phospholipid antibodies: negative
Other	
	*FBC: mild lymphopenia*
	ESR, U&Es, LFTs: normal
	Hepatitis B serology–previous immunisation
	Hepatitis C serology: negative.

	Thyroid function: normal
	*Thyroid autoantibodies: raised (458 U/mL).*

	VZV IgM negative, VZV IgG positive, HSV 1&2 IgG and IgM: negative.

**Table 8 brainsci-12-00306-t008:** Baseline characteristics, audiovestibular dysfunction and diagnoses in the three cases presented.

	Case 1	Case 2	Case 3
Gender	Male	Male	Female
Ethnicity	White British	Black Ghanaian	White British
Age at presentation	28	17	46
Occupation	Carpenter	Student	Medical doctor
Pre-existing medical conditions	Ulcerative colitis, migraine	Prior episode of uveitis age 4	Allergic rhinitis/atopy
Hearing loss	Asymmetric, fluctuant SNHL	Insidious right SNHL, then sudden onset bilateral profound hearing loss	Asymmetric, fluctuant, high-frequency SNHL
Vestibulopathy	Subjective: episodic vertigo, later imbalance Objective: bilateral vestibulopathy	Subjective: sudden onset imbalance Objective: bilateral vestibulopathy	Subjective: sudden onset imbalance Objective: bilateral vestibulopathy
Diagnosis	Secondary AIED (in context of ulcerative colitis)	Cogan syndrome	Possible primary AIED

## Data Availability

Not applicable.

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
