# Peer review of "Autoimmune Vestibulopathy—A Case Series"

_brainsci, 2022, doi:10.3390/brainsci12030306_

Round 1

Reviewer 1 Report

This interesting and informative case report presents three different clinical presentations of audiological and vestibular symptoms in systemic autoimmune diseases. Although reported in the form of case reports due to the rarity of the autoimmune inner ear disease, the manuscript is well-written. I have reviewed it thoroughly and have no further comments.

Author Response

Dear Reviewer 1:

Thank you very much for your insightful comments and views. We have taken your suggestions on board and made some minor amendments. We warmly welcome your opinions on these revisions.

Reviewer 1 comments:

This interesting and informative case report presents three different clinical presentations of audiological and vestibular symptoms in systemic autoimmune diseases. Although reported in the form of case reports due to the rarity of the autoimmune inner ear disease, the manuscript is well-written. I have reviewed it thoroughly and have no further comments.

Response: We are grateful to the Reviewer for the positive remarks.

Finally, in relation to methodological concerns from both reviewers, we have included the following statement in the Discussion (page 16, lines 417 - 421) to justify the non-standardisation of investigations across the three cases:

“The absence of predefined management guidelines or pathways for AIED led to a lack of standardised investigations in these three cases herein described. We are developing a department-specific AIED algorithm to address this shortfall, with a view to National and International validation.”

We look forward to hearing your thoughts once again.

Reviewer 2 Report

The authors wrote a case series about Autoimmune vestibulopathy. They presented 3 cases, the last one related to Pfizer vaccination. The article is very interesting, well written and the topic is always hot, because there are not guidelines about autoimmune inner ear disease and the therapy is always different, changing case for case. The article is worthy of publication. It is quite long for a case series, but I liked every single presentation, with a detailed audiological and vestibular description and interesting MRI images. To improve the scientific impact, I can give some suggestions to add in the discussion.

You used VHIT to value the vestibular function correctly, in the last case you used also cVEMPs and caloric test. I'd like suggest to report more about the possible use of the cVEMPs in hearing loss, using this reference: Ciodaro F, Freni F, Alberti G, Forelli M, Gazia F, Bruno R, Sherdell EP, Galletti B, Galletti F. Application of Cervical Vestibular-Evoked Myogenic Potentials in Adults with Moderate to Profound Sensorineural Hearing Loss: A Preliminary Study. Int Arch Otorhinolaryngol. 2020 Jan;24(1):e5-e10. 

An important question, introduced recently, is the possible use of cochlear-vestibular implant to treat the hearing loss and the vestibular impairment. Do you think that in the auto-immune disease should be useful ? Please use these references:

Freni F, Gazia F, Slavutsky V, Scherdel EP, Nicenboim L, Posada R, Portelli D, Galletti B, Galletti F. Cochlear Implant Surgery: Endomeatal Approach versus Posterior Tympanotomy. Int J Environ Res Public Health. 2020 Jun 12;17(12):4187. 

Rubinstein JT, Ling L, Nowack A, Nie K, Phillips JO. Results From a Second-Generation Vestibular Implant in Human Subjects: Diagnosis May Impact Electrical Sensitivity of Vestibular Afferents. Otol Neurotol. 2020 Jan;41(1):68-77.

Author Response

Dear Reviewer 2

Thank you very much for your insightful comments and views. We have taken your suggestions on board and made some minor amendments. We warmly welcome your opinions on these revisions.

Reviewer 2 comments:

The authors wrote a case series about Autoimmune vestibulopathy. They presented 3 cases, the last one related to Pfizer vaccination. The article is very interesting, well written and the topic is always hot, because there are not guidelines about autoimmune inner ear disease and the therapy is always different, changing case for case. The article is worthy of publication. It is quite long for a case series, but I liked every single presentation, with a detailed audiological and vestibular description and interesting MRI images. To improve the scientific impact, I can give some suggestions to add in the discussion.

You used VHIT to value the vestibular function correctly, in the last case you used also cVEMPs and caloric test. I'd like suggest to report more about the possible use of the cVEMPs in hearing loss, using this reference: Ciodaro F, Freni F, Alberti G, Forelli M, Gazia F, Bruno R, Sherdell EP, Galletti B, Galletti F. Application of Cervical Vestibular-Evoked Myogenic Potentials in Adults with Moderate to Profound Sensorineural Hearing Loss: A Preliminary Study. Int Arch Otorhinolaryngol. 2020 Jan;24(1):e5-e10.

Response: We are grateful to the Reviewer for the comments and suggestions. We have expanded the Discussion section (page 16, lines 421 - 429) to reflect this comment and have included the suggested reference:

“vHIT was the vestibular function test of choice in all three cases in this series but, as seen in case 3, a small battery of vestibular investigations may better evaluate the VOR at different frequencies (e.g. high frequency 4-7 Hz vHIT vs. low frequency 0.002 – 0.004 Hz caloric) and facilitate anatomical localisation of a peripheral vestibular lesion (evaluation of saccular, utricle and semicircular canal function or superior vs. inferior vestibular nerve involvement). Impulsive rotation and caloric tests are the more ‘traditional’ vestibular investigations but cVEMPs are increasingly used to non-invasively evaluate labyrinthine function, specifically that of the saccule and inferior vestibular nerve, in cases of moderate to profound sensorineural hearing loss [62].”

An important question, introduced recently, is the possible use of cochlear-vestibular implant to treat the hearing loss and the vestibular impairment. Do you think that in the auto-immune disease should be useful ? Please use these references:

Freni F, Gazia F, Slavutsky V, Scherdel EP, Nicenboim L, Posada R, Portelli D, Galletti B, Galletti F. Cochlear Implant Surgery: Endomeatal Approach versus Posterior Tympanotomy. Int J Environ Res Public Health. 2020 Jun 12;17(12):4187.

Rubinstein JT, Ling L, Nowack A, Nie K, Phillips JO. Results From a Second-Generation Vestibular Implant in Human Subjects: Diagnosis May Impact Electrical Sensitivity of Vestibular Afferents. Otol Neurotol. 2020 Jan;41(1):68-77.

Response: We are once again grateful to the Reviewer for raising this important point. We have amended the Discussion section (page 17, lines 467 - 476) to include statements regarding vestibulo-cochlear implants in AIED, and included the second suggested reference:

“In cases where hearing deteriorates despite immunomodulatory therapy, cochlear implantation is a viable option often resulting in excellent outcomes [70], although ossification can affect up to 50% of patients receiving long-term treatment for AIED [71] and therefore early implantation is desirable to avoid traumatic insertion associated with post-inflammatory cochlear obliterative changes. Rubenstein et al. recently described generation of vestibular electrically evoked compound action potentials following placement of a vestibular-cochlear implant in two patients with bilateral Meniere’s disease and bilateral vestibular loss [72], and therefore vestibular-cochlear implantation may also emerge as a potential treatment option for AIED in the future.”

Finally, in relation to methodological concerns from both reviewers, we have included the following statement in the Discussion (page 16, lines 417 - 421) to justify the non-standardisation of investigations across the three cases:

“The absence of predefined management guidelines or pathways for AIED led to a lack of standardised investigations in these three cases herein described. We are developing a department-specific AIED algorithm to address this shortfall, with a view to National and International validation.”

We look forward to hearing your thoughts once again.